# Food Reward after a Traditional Inuit or a Westernised Diet in an Inuit Population in Greenland

**DOI:** 10.3390/nu14030561

**Published:** 2022-01-27

**Authors:** Hanne Pedersen, Kristine Beaulieu, Graham Finlayson, Kristine Færch, Marit Eika Jørgensen, Jack Ivor Lewis, Mads Vendelbo Lind, Lotte Lauritzen, Jonas Salling Quist

**Affiliations:** 1Clinical Research, Copenhagen University Hospital—Steno Diabetes Center Copenahgen, Borgmester Ib Juuls Vej 83, DK-2730 Herlev, Denmark; k.beaulieu@leeds.ac.uk (K.B.); g.s.finlayson@leeds.ac.uk (G.F.); kristine.faerch@regionh.dk (K.F.); jonas.salling.quist@regionh.dk (J.S.Q.); 2National Institute of Public Health, University of Southern Denmark, Studiestræde 6, DK-1455 Copenhagen, Denmark; maej@peqqik.gl; 3Appetite Control and Energy Balance Research, School of Psychology, University of Leeds, Leeds LS2 9JT, UK; 4Department of Biomedical Sciences, University of Copenhagen, DK-2200 Copenhagen, Denmark; 5Clinical Epidemiology, Steno Diabetes Center Copenhagen, Borgmester Ib Juuls Vej 83, DK-2730 Herlev, Denmark; 6Steno Diabetes Center Greenland, Dronning Ingrids Vej, Nuuk 3900, Greenland; 7Department of Nutrition, Exercise and Sports, University of Copenhagen, DK-2200 Copenhagen N, Denmark; jack.lewis@nexs.ku.dk (J.I.L.); mads_lind@hotmail.com (M.V.L.); ll@nexs.ku.dk (L.L.)

**Keywords:** food reward, wanting, liking, Inuit, diet, food intake

## Abstract

The food availability and dietary behaviours in Greenland have changed with increasing Westernisation. Food reward is an important driver of food choice and intake, which has not previously been explored in the Arctic population. The aim of this study was to explore differences in food reward after a four-week intervention period with a traditional Inuit diet (TID) or Westernised diet (WD) in Inuit populations in Northern and Western Greenland. This cross-sectional analysis included 44 adults (*n* = 20 after TID and *n* = 24 after WD). We assessed the food reward components, explicit liking and implicit wanting, using the Leeds Food Preference Questionnaire under standardised conditions 60 min after drinking a glucose drink as part of an oral glucose tolerance test after four weeks following a TID or WD. The food intake was assessed using food frequency questionnaires. The intervention groups differed only in implicit wanting for high-fat sweet foods, with higher implicit wanting among the participants following TID compared to WD. Both groups had lower explicit liking and implicit wanting for sweet relative to savoury foods and for high-fat relative to low-fat foods. This exploratory study can guide future studies in Inuit populations to include measures of food reward to better understand food intake in the Arctic.

## 1. Introduction

The Arctic Inuit populations have survived in a food landscape characterised by limited arable land and a short summer growing season. These populations have primarily been dependent on fishing and hunting wild animals. However, with increased Westernisation, the food environment and, thus, food choices have changed in conjunction with changes in health outcomes, such as increasing prevalence of obesity. Yet, the changes in food-related behaviour among the Arctic Inuit populations are not well established and need further investigation.

The traditional foods of the Greenlandic diet include local fish and other sea food (e.g., salmon, halibut, and trout), marine mammals (e.g., seal, whale, and walrus), wild terrestrial animals (e.g., caribou and musk ox), and game birds (e.g., guillemot, and eider ducks and their eggs) [1]. Historically, the availability of European foods has been limited due to both policy restrictions and practical challenges concerning shipping and distribution [2]. However, since the middle of the 20th century, progressive industrialisation has increased Western, and primarily Danish, food availability in Greenland and changed dietary behaviours, leading to a decreased consumption of traditional foods [3]. Data from a country-wide health survey from 2005 to 2007 report that this impact is more pronounced in towns compared to villages and in younger generations compared to the elderly [4]. A large cross-sectional study with data collected between 2005 and 2010 from 2752 Inuit adults assessed food intake using food frequency questionnaires (FFQ) and found that traditional foods contributed to 21% of the energy intake [1], in accordance with previous studies [5,6]. The study also found that people with a higher intake of traditional foods had a lower intake of sugar and fibre and a tendency towards a lower intake of saturated fat and an overall healthier dietary fat profile [1]. The authors, therefore, recommended increased consumption of traditional food alongside healthy fibre-dense and low-fat imported foods [1] in line with the Greenland Board of Nutrition [7]. Local foods have also been promoted as a strategy for sustainable community development, providing economic possibilities for local hunters and reducing imports [2]. Alongside the recommendations of a traditional Inuit diet (TID), traditional foods are viewed as healthy and highly appreciated among Greenlanders and further manifest a valuable connection to the Greenlandic culture [3]. A dietary questionnaire from 2018 showed that only 10% of the people in settlements and 18% in towns followed at least four of the five dietary recommendations, e.g., 39% consumed fruits and 24% consumed vegetables daily, 43% consumed fish at least once a week, and 44% had a daily intake of sugar-sweetened beverages. Moreover, the data showed an overall increase in the consumption of vegetables but also of sugar-sweetened beverages as well as a decrease in the intake of fish [8].

The decreased intake of traditional foods in favour of imported foods may partly be explained by a wish to vary the diet as well as difficulties in obtaining the traditional foods and the associated costs [3]. However, the food choices and behaviours behind these dietary changes are driven by a complex interplay between a variety of external and internal factors [9]. An important driver of food choice and intake is food reward, which has been associated with both overeating and obesity [10,11]. Food reward is characterised by the components ‘liking’ and ‘wanting’ [10]. Liking is described as the pleasure derived from food, whereas wanting refers to the motivation for a rewarding food triggered by cues, such as the sight or smell of food [10]. These aspects of food reward can be measured explicitly and implicitly using the Leeds Food Preference Questionnaire (LFPQ) [12]. The components of food reward in Arctic Inuit populations have not previously been explored but may increase our understanding of why this population gradually consumes more imported foods. In particular, some studies in animals and a few human studies suggest that a Western diet with energy-dense foods can modulate the reward for these same foods [13,14]. Moreover, the rewarding aspects of highly palatable foods, including high-fat sweet foods, have been associated with increased intake [9]. Compared to Western culture, the introduction of the ultra-processed high-fat and sweet foods has been fast and recent. Such a trend could potentially affect diet-related obesity, diabetes, and cardiovascular disease in the Arctic Inuit populations. This is relevant with the high prevalence of obesity (28%), diabetes (9%), and cardiovascular diseases (10–13%) in Greenland [8,15,16]. As part of the largest interventional study on the effects of a TID on glucose homeostasis and metabolic health markers, it was made possible to study behavioural aspects of food reward in the current study.

On this background, we aimed to explore differences in food reward after four weeks following either a TID or a Westernised diet (WD) in an Inuit population in Greenland and to explore the implementation of the method to assess food reward in this population. Food reward was assessed in response to food cues that are typical for a Western diet. More specifically, we examined preferences for foods with a sweet relative to savoury taste, and for foods with high relative to low fat content as well as preferences for combined food categories differing in taste and fat content. Moreover, the food intake before and during the intervention was described to characterise the diet groups.

## 2. Materials and Methods

### 2.1. Study Population and Location

This present exploratory study included a subgroup of 44 participants from a clinical randomised cross-over trial that examined whether a TID compared to a WD could improve glycaemic control in Greenland Inuit adults. This analysis included only participants who performed the LFPQ after the first intervention period. The trial is registered at ClinicalTrials.gov (NCT04011904), approved by the Ethics Committee of Greenland (KVUG 2018-26), and described in detail elsewhere [17]. Out of three study locations (Nuuk, Qaanaaq, and Qasigiannguit) in Greenland, this subgroup included participants from Qaanaaq and Qasigiannguit in north-western and western Greenland, respectively. Participants were 18–80 years old and had a BMI ≥ 18.5 kg/m^2^. Participants were excluded if they were diagnosed with or pharmacologically treated for diabetes, had a history of severe hypertriglyceridemia, or used systemic peroral glucocorticoids or injected steroids. Data were collected from September 2019 to March 2020.

### 2.2. Intervention for the Present Subgroup

This sub-study comprised one four-week dietary intervention period in which the 44 participants followed either the TID or WD (Figure 1). Food reward was assessed on a test day after the intervention period. No baseline measurements of food reward were conducted as this outcome was added subsequently to the original study design and it was considered too burdensome for participants to add more measurements on their first test day. Moreover, it was not possible to measure food reward on the majority of participants after the second intervention period of the main cross-over trial due to the first lockdown period of COVID-19. During the intervention period, the participants were provided at least 20% of their total energy intake with foods from local supermarkets. The foods provided during the TID intervention consisted of fish, marine mammals, shellfish, and traditional meats, and participants were advised to reduce their intake of grains, fast food, and other imported foods. The diet was targeted to be high in fat (>40% of the total energy intake (E%)) and low in carbohydrates (<30 E%). The WD was targeted to have a high content of carbohydrates (55–65 E%) and low-fat content (30–35 E%), and the provided foods consisted of cereal products and imported meats and poultry. On test days, participants were instructed how to incorporate the dietary changes into their daily life by trained staff and in a written manual. In both dietary interventions, the participants were recommended to limit their alcohol intake, and none of the interventions had specific recommendations regarding fruits and vegetables.

### 2.3. Study Procedure and Examinations

#### 2.3.1. Food Reward

The food reward measures, liking and wanting, were assessed after the intervention using the LFPQ [12]. Food reward was measured approximately one hour into an oral glucose tolerance test (OGTT). Due to the sweet sensory properties of the OGTT, it was expected that reward responses for sweet foods would be suppressed. Nevertheless, this controlled context provided a standardised reference point from which differences between TID and WD could be compared. The computerised LFPQ assessed participants’ liking and wanting for 16 ready-to-eat foods (Figure 2). Before the study, the LFPQ was translated into Greenlandic by study staff with a social health background who had Greenlandic as first and Danish as second language. Food items included in the LFPQ differed in the content of fat (high or low) and taste (sweet or savoury), creating four combined food categories: high-fat savoury foods, low-fat savoury foods, high-fat sweet foods, and low-fat sweet foods. The foods were displayed on a computer screen as images that were validated in a Danish context [18]. All foods were considered appropriate for breakfast/brunch and assumed to be well-recognised and overall liked in a Greenlandic setting where most of the imported foods are shipped from Denmark, although some of the items are rarely available in Greenland.

Before starting the task, participants were introduced to all food images and instructed in the task verbally and in writing in Greenlandic. The LFPQ consisted of two parts. In the first part, participants rated their explicit liking for each of the 16 foods on a 100-point visual analogue scale. Foods were shown as images together with a question at the top of the screen: “How pleasant would it be to taste some of this food now?” The question was answered by clicking with a mouse on the place on the scale that matched the participant’s perception in that given moment. The second part was a forced choice paradigm where participants chose between two images of foods from different food categories, presented at the same time, by answering the question: “Which food do you most want to eat now?” This part consisted of a total of 96 choices that participants were asked to make as fast as possible by pressing one of two keys on a keyboard.

#### 2.3.2. Anthropometry

At baseline and after the intervention, body weight was assessed to the nearest 100 g using a scale from which body composition was also analysed using bioelectrical impedance (Tanita TBF-300MA (Tanita Corporation, Tokyo, Japan)). Height was measured at baseline to the nearest centimetre.

#### 2.3.3. Interviews and Assessment of Food Intake

Interviews were conducted after initiating the OGTT at baseline and at the end of the intervention period. During interviews, the trained study staff obtained information about socioeconomic status (at baseline) as well as alcohol and food intake. Food intake was assessed using a 44-item FFQ [19] and reflecting habitual diet and intervention diets (TID or WD) over the preceding 4 weeks, respectively. During the interview, participants were asked about the frequency (day/week/month/year) and estimated portion size (amounts/slices/glasses/bottles) of foods consumed within different food groups, including both traditional and imported foods.

### 2.4. Data Analyses & Calculations

#### 2.4.1. Food Reward

##### Sweet Bias and Fat Bias Scores

The scores that summarised results for the four combined LFPQ food categories (see below: Combined Food Categories) were used to calculate sweet bias and fat bias for both explicit liking and implicit wanting [12]. To calculate sweet bias, mean scores for savoury foods were subtracted from mean scores for sweet foods, e.g., explicit liking scores from ratings of savoury foods were subtracted from explicit liking scores from ratings of sweet foods (and similar for implicit wanting scores). Similarly, to calculate fat bias, mean scores for low-fat foods were subtracted from mean scores for high-fat foods. This provided bias scores such that positive scores for sweet bias indicated higher preference for sweet relative to savoury foods, and positive scores for fat bias indicated higher preference for high-fat relative to low-fat foods. In contrast, negative scores for sweet bias and fat bias reflected higher preference for the other taste (savoury foods) or fat content (low-fat foods), respectively.

##### Combined Food Categories

To assess explicit liking for the four combined food categories, the average rating from 0–100 on the visual analogue scale was calculated for all four foods within a food category. Implicit wanting for each of the four combined food categories was calculated based on a combination of reaction time and choice or non-choice of foods in the forced choice paradigm [12]. An ‘implicit wanting’ score for a specific food category that was above zero indicated a higher preference for this food category compared to other food categories, and a score below zero indicated a lower preference for that particular food category compared to the other food categories.

#### 2.4.2. Food Intake

Frequency and portion size of foods consumed within each food category of the FFQ were converted to grams per day [19]. Moreover, intake of specific foods were grouped into 14 categories: marine mammals (seal, whale, and mattak); fish (cod, halibut, ammassat, trout/salmon, fish cold cut, and other fish); traditional meat (reindeer/musk, game bird, and dried fish or meat); berries; imported meat (beef, pork, lamb, poultry, and cold cut); fruit/fruit juice (apple/pear/bananas, orange/grapefruit, other fruit, and fruit juice); vegetables (mixed vegetables, carrot, cruciferous vegetables, potatoes, and tomato); dairy products (milk and cheese); cereal products (rye bread, French bread, oats, breakfast cereal, pasta, and rice); cake; candy (sweets/chocolate); sugar-sweetened beverages (soda and squash drink); sugar in coffee/tea; and ultra-processed high-fat foods (pizza/burger, fries, and crisps).

### 2.5. Statistical Methods

Data were checked for normality, and descriptive data are presented as mean (SD) for normally distributed data and as median (IQR (interquartile range)) for non-normally distributed data. As food reward and food intake variables were non-normally distributed, they were described using median (IQR) scores. To examine differences in food reward scores between participants who received the TID or WD intervention, a general linear model adjusted for sex and age was used. Due to restrictions mentioned above, food reward was only measured after the intervention period. However, as participants were randomly allocated to diet groups, we do not assume any baseline imbalances. Food intake at baseline and during the intervention was stratified by diet (TID or WD). This study was exploratory, and it is uncertain whether the sample size was sufficient to detect differences between diet groups. However, according to studies with a similar sample size (*n* = 44), it is possible to detect differences between groups in both explicit liking and, to some extent, implicit wanting in both cross-sectional [20,21] and interventional studies [22,23,24,25].

## 3. Results

### 3.1. Participant Characteristics

In total, 44 participants were studied, of whom 20 followed a TID and 24 followed a WD for four weeks prior to examining food reward and intake. The participant characteristics are described in Table 1. Out of the 47 participants completing the first two visits at the two study sites, three participants from the TID group did not complete the LFPQ and were excluded from the study. Moreover, one WD participant was excluded from the analyses of implicit wanting because the reaction times for choosing an image were <100 ms, which is implausible for visually processing and choosing (pressing the keyboard) an image.

### 3.2. Food Reward

#### 3.2.1. Sweet Bias and Fat Bias Scores

Both food reward components, explicit liking and implicit wanting, were assessed in relation to participants’ preferences for sweet over savoury (sweet bias) and for high-fat over low-fat (fat bias). No overall differences were found between the diet groups (TID and WD) for either sweet or fat bias after the intervention. Overall, the sweet bias results indicated that the participants had a lower explicit liking (TID: −10.6 (−15.8, −1.1); WD: −11.6 (−21.6, −4.8)) and implicit wanting (TID: −18.4 (−45.2, 1.4); WD: −21.8 (−40.0, −10.6)) for sweet foods relative to savoury foods (Figure 3 (top) and Appendix A). The fat bias results indicated that the participants had a lower explicit liking (TID: −15.9 (−21.9, −3.7); WD: −24.4 (−30.4, −2.2)) and implicit wanting (TID: −19.7 (−30.4, −6.6); WD: −31.3 (−50.0, −17.0)) for high-fat foods relative to low-fat foods (Figure 3 (bottom) and Appendix A).

#### 3.2.2. Combined Food Categories

When dividing the results into the four combined food categories (high-fat sweet, low-fat sweet, high-fat savoury, and low-fat savoury foods), implicit wanting for high-fat sweet foods was higher after four weeks on the TID compared to WD (TID: −38.7 (−52.7, −32.6) vs. WD: −52.4 (−58.6, −43.1); *p* = 0.029) (Figure 4 and Appendix A). No other differences in food reward were found between the diet groups. For both diet groups, the results indicate that the participants had lower explicit liking and implicit wanting for foods characterised as high in fat with sweet taste compared to the other three food categories (Figure 4 and Appendix A).

### 3.3. Food Intake

The intake (grams/day) of different types of foods was summarised for each dietary intervention group (TID and WD) as assessed by the FFQ on test days at the baseline and after the intervention (Table 2). At the baseline, both dietary intervention groups had a higher intake of imported compared to traditional Inuit foods but a relatively low intake of ultra-processed, imported foods except for sugar-sweetened beverages. As expected, the results indicate that the participants were compliant to the assigned diets, i.e., those who were in the TID group increased their intake of traditional Inuit foods, whereas the participants in WD decreased their intake of these foods. Moreover, the TID group numerically decreased their intake of imported foods, particularly meat, fruit/fruit juice, vegetables, dairy products, cereal products, and sugar-sweetened beverages. The WD group increased numerically only with respect to their intake of meat and dairy products.

## 4. Discussion

This study presented a unique opportunity to collect exploratory data on food reward in an Inuit population. Food reward did not differ between the groups following a TID or a WD for four weeks up to the assessment, except for the participants in the TID group having higher implicit wanting for high-fat sweet foods compared to the participants in the WD group. Using standardised measures following OGTT, we showed in this context that participants’ overall explicit liking and implicit wanting were lower for sweet relative to savoury foods and for high-fat relative to low-fat foods after following either a TID or a WD for four weeks. When food reward was summarised into the four food categories of the LFPQ, the preferences for high-fat sweet foods appeared to be lower relative to the three other food categories for both diet groups. Food intake at the baseline was characterised as primarily consisting of imported foods. During the four weeks of dietary intervention, the TID group increased their intake of traditional foods, while the WD group slightly increased their intake of imported foods in accordance with their dietary group assignment.

Food reward, as measured in the LFPQ, is thought to be momentary [26] and can, therefore, be affected by several external and internal factors, such as the current status of hunger, prior food intake, time of day, social interactions, etc. [27]. Moreover, genetic, metabolic, and psychological factors are all expected to affect food reward [28,29]. Cultural differences in hedonic responses to sweet and creamy solutions have been found in populations such as Pima Indians and Whites [30], which add to the interest in examining food reward in an Inuit population, who traditionally had a monotonous diet but now have an increasing availability of a variety of foods [31]. When comparing across food categories, our findings indicate that the Inuit population displays similar trends in food reward to a Danish adult population with normal weight [18]. Within categories, however, this Inuit population had a numerically lower liking and wanting for high-fat sweet and low-fat sweet foods compared to the Danish population. This contradicts prior studies indicating a high intake of (and probably preference for) sugars in the Inuit population. A study among Inuit adults in Greenland found the intake of added sugar to be above the recommendations for 71% of the men and 67% of the women [1]. Moreover, sugar added to coffee/tea alone contributed to 6 E%, and carbonated soft drinks contributed to 5 E% in that population. In this sub-study, the participants had a low intake of sugar added to coffee/tea but a high intake of sugar-sweetened beverages at the baseline and after the WD intervention. However, in this study, it is highly likely that the sweet glucose solution in the OGTT, which participants consumed one hour before performing the LFPQ, explains their lower preference for sweet tasting foods. The low-fat sweet category (e.g., fruits) was probably lower in glucose compared to the high-fat sweet category (e.g., brownies, muffins), which could partly explain the higher preference for low-fat compared to high-fat sweet foods. Indeed, sweet preloads have been found to modulate preferences away from sweet food to similar preferences for sweet and savoury foods [32].

The participants in the two groups differed only in their implicit wanting for high-fat sweet foods. We must consider the risk of a false positive finding due to multiple testing or the difference being driven by a few outliers in the TID group, but we also acknowledge the possibility that a transition from a mixed diet to TID, which was observed in the changes in food intake during this study, could have increased the desire for high-fat sweet foods that are not part of the TID. Considering the literature suggesting that a WD including high-energy dense foods can increase reward for high-fat sweet foods [13,14], our finding is counterintuitive and should be further explored in future studies. Changes in weight would also be a possible explanation for the differences in food reward between groups but is not relevant in this study as none of the groups appeared to change their body weight or body composition significantly during the intervention period.

During this study, the participants were exposed to images of Western foods that were assumed to be well-recognised and generally liked among the Greenlandic population. However, some foods may not have been generally available at the study sites to the extent expected prior to initiating the measurements. This could have affected participants’ ratings and choices in the task. Qaanaaq is the northernmost city, receiving food shipments only two times per year and with limited possibilities of import by plane (one flight per week). Overall, the availability of fresh foods, such as melon, cucumber, berries, fruits, and salad, is limited in the Arctic due to high costs and short shelf life [33]. It is possible that some of the fresh food products could have been rated higher because of their exclusivity due to low availability.

The transition towards Western food choices and intake can be caused by multiple factors [3], and the consequences of this transition are still to be established. However, the increasingly prevalent Western diet is characterised by abundance and high energy density with processed foods high in fat and sugar and is associated with increased levels of obesity and non-communicable diseases [34,35]. Moreover, a high prevalence of obesity, diabetes, and certain types of cardiovascular disease has been observed in the Inuit population during the last few decades [8,15,16,36]. However, the increased availability of Western foods also includes vegetables, dairy products, imported meats, and high-fibre bread that could potentially add to a more varied and healthy diet. A study in a Greenlandic population rated the following foods as most preferred: rye bread, potatoes, vegetables, apples, and lamb [3]. Food reward studies in this population would enable us to examine whether certain population groups are more susceptible to the palatable and abundant Western food environment, including high-fat sweet foods, which can result in consequent unhealthy weight gain.

The strengths of this study include the novelty of examining food reward in a remote area of the world as part of the first and largest interventional study on the effects of a TID on glucose homeostasis and metabolic health markers [17]. Conducting health-related studies in this population is challenging due to the infrastructure across the country, limited access to health care resources, technology, and devices, as well as language and cultural barriers, which makes this a unique opportune dataset to study. However, this study also has limitations that must be highlighted to provide guidance for future studies on food reward in the Arctic Inuit population. First, the foods included in the LFPQ were primarily imported foods; hence, some of them may not have been frequently consumed, at least not throughout the year, as winter introduces many challenges of importing and distributing products throughout the country. Future studies should culturally adapt images in the LFPQ to the Greenlandic population, and the Greenlandic LFPQ should be back translated. Traditional food items should be included in the LFPQ to examine differences between responses to Western and traditional Inuit foods. Western foods should be selected according to the current availability and frequency of consumption in different areas of Greenland and could be selected by collaborating with local grocery store managers around the country. This was the first time conducting the LFPQ in a Greenlandic setting and, because of language barriers, it was uncertain whether all the participants understood the task. Challenges with being able to read and understand instructions, and potentially poor eye sight [37], should also be assessed more systematically and considered in future studies. As we do not have baseline measurements of food reward, we could only assume an equal distribution of the food reward scores at the baseline. This was reasonable as the participants were randomly allocated to a diet and we do not expect differences in the results to be due to other confounding factors. Therefore, we expect differences between the groups after the intervention period to be explained by the exposure to the two diets. Another concern is that the small sample size could increase the risk of a type 2 error. Therefore, future studies should ensure a sufficient sample size and, moreover, include repeated measures (at least baseline and post-intervention) for a stronger study design. Furthermore, the potential effects of the OGTT on food reward makes it impossible to generalise the findings to other settings in the Arctic Inuit population but could be avoided in future studies by ensuring a standardized mixed meal or overnight fast prior to testing as well as controlling for hunger status. Food intake was assessed by an FFQ, which aims to assess intake over a longer period, but the intake estimates were reported in food categories and based on recalls during interviews, limiting the certainty of the data [38]. Lastly, this study measured food reward only in two smaller towns in north-western and western Greenland, Qaanaaq and Qasigiannguit, which limits the generalisability to larger towns and smaller settlements with a different food availability and socio-economic status [33,39]. Future studies of food reward should be conducted in different areas of Greenland that take into account the differences in food availability and variety as well as different age groups. The relative high age of the population in this sub-study compared to the inclusion criteria could be due to the participants being partly recruited based on prior participation in population studies conducted several years back. Younger generations are raised in a fast-moving, globalised world, which may affect their relation to food differently than older generations. Understanding the effects of changes in the food environment on food choices, intake, and behaviour in early life can inform future strategies and recommendations to promote healthy lifestyles in Greenland for the generations to come.

## 5. Conclusions

In conclusion, these first results on food reward studied in an Arctic Inuit population show no differences between the groups after following either a four-week TID or WD, except for higher implicit wanting for high-fat sweet foods among those following a TID. Despite the circumstances and limitations for measuring food reward in this sample, this study hopefully sets the scene for measuring food choices and eating behaviour change, including food reward, in future studies in the Inuit population to better establish the impact of the traditional Inuit diet versus the rapidly changing food environment towards a more Westernised diet.

## Figures and Tables

**Figure 1 nutrients-14-00561-f001:**
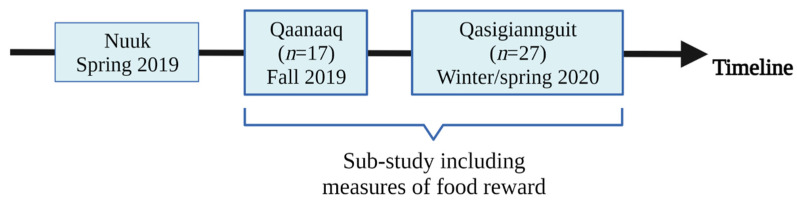
Timeline displaying when and where data on food reward were collected during the original study. Created with BioRender.com (last accessed 24 January 2022).

**Figure 2 nutrients-14-00561-f002:**
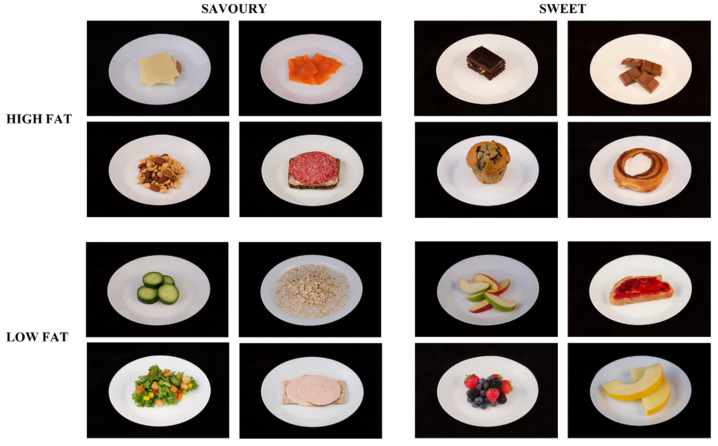
Food images included in the Leeds Food Preference Questionnaire. Foods differed according to sweet and savoury taste and fat content, resulting in four combined food categories.

**Figure 3 nutrients-14-00561-f003:**
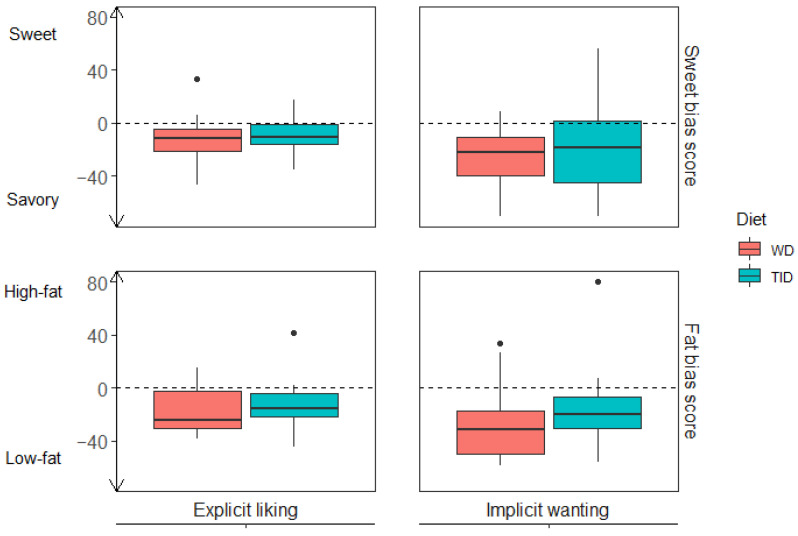
Sweet and fat bias scores stratified by diet group (TID and WD). Upper panel (Sweet bias): explicit liking (left, *n*= 44) and implicit wanting (right, *n* = 43) for sweet relative to savoury foods where scores below zero indicate preferences for savoury over sweet. Lower panel (fat bias): explicit liking (left, *n* = 44) and implicit wanting (right, *n* = 43) for high-fat relative to low-fat foods where scores below zero indicate preferences for low-fat over high-fat. Coloured boxes represent median and IQR, whiskers represent Q − 1.5*IQR and Q3 + 1.5*IQR, and dots represent outliers. WD, Westernised diet; TID, traditional Inuit diet; IQR, interquartile range.

**Figure 4 nutrients-14-00561-f004:**
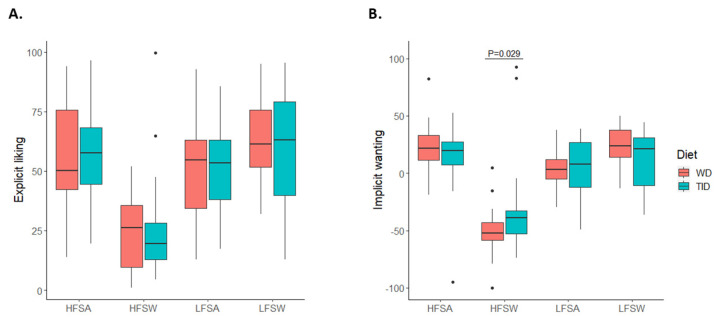
Liking and wanting for different food categories stratified by diet. Explicit liking (**A**, *n* = 44) and implicit wanting (**B**, *n* = 43) for the four combined food categories: high-fat savoury foods (HFSA), low-fat savoury foods (LFSA), high-fat sweet foods (HFSW), and low-fat sweet foods (LFSW). Coloured boxes represent median and IQR, whiskers represent Q1−1.5*IQR and Q3 + 1.5*IQR, and dots represent outliers. WD, Westernised diet; TID, traditional Inuit diet; IQR, interquartile range.

**Table 1 nutrients-14-00561-t001:** Participants’ characteristics at baseline (visit 1) and after the intervention (visit 2) (*n* = 44).

	Traditional Diet		Westernised Diet	
	Visit 1	Visit 2	Visit 1	Visit 2
*n*	20	20	24	24
Genotype = Homozygous carriers	3 (15.0)		1 (4.5)	
Place of residence (Qaanaaq/Qasigiannguit)	10/10		7/17	
Sex = Male	10 (50.0)		11 (45.8)	
Age, years	60.5 (11.7)		55.4 (9.5)	
Weight, kg	67.7 (17.3)	66.6 (16.8)	76.1 (15.6)	76.1 (15.5)
BMI, kg/m^2^	26.0 (5.9)	25.6 (5.7)	27.9 (5.0)	27.9 (4.7)
Fat, %	28.3 (10.1)	27.8 (9.8)	32.1 (10.4)	31.9 (10.0)
Fat-free mass, kg	47.7 (11.1)	47.3 (10.8)	51.2 (11.0)	51.4 (10.9)
Alcohol frequency				
More than 2 times per week	3 (15.0)	1 (10.0)	1 (4.2)	1 (5.3)
2 times per month or less	17 (85.0)	9 (90.0)	23 (95.8)	18 (94.7)
Weekly alcohol intake				
0 units	5 (25.0)	6 (30.0)	8 (33.3)	10 (41.7)
1–7 units	3 (15.0)	2 (10.0)	5 (20.8)	0 (0.0)
8–14 units	3 (15.0)	2 (10.0)	3 (12.5)	4 (16.7)
15 or more units	2 (10.0)	1 (5.0)	0 (0.0)	0 (0.0)
Missing	7 (35.0)	9 (45.0)	8 (33.3)	10 (41.7)
Smoking status				
Current smoker	12 (60.0)		14 (58.3)	
Previous smoker	8 (40.0)		6 (25.0)	
Never smoked	0 (0.0)		4 (16.7)	
Educational level				
8th grade or less	8 (40.0)		8 (33.3)	
9th to 12th grade	12 (60.0)		16 (66.7)	
Employment				
Full-time paid	10 (50.0)		16 (66.7)	
Part-time paid	1 (5.0)		2 (8.3)	
Self-employed (fishing/fisheries)	0 (0.0)		1 (4.2)	
Pensioner	7 (35.0)		3 (12.5)	
Other	2 (10.0)		2 (8.3)	

Values are presented as mean (SD) for continuous variables and number (%) for categorical variables. Anthropometric characteristics and alcohol consumption were assessed during both visit 1 and visit 2.

**Table 2 nutrients-14-00561-t002:** Food intake (grams/day) at baseline and during the intervention (*n* = 44).

	Traditional Inuit Diet		Westernised Diet	
	Baseline	During Intervention	Baseline	During Intervention
*n*	20	20	24	24
Traditional foods				
Berries	3.0 (0.2, 12.9)	0.0 (0.0, 9.9)	3.0 (0.7, 4.3)	0.7 (0.0, 3.0)
Marine mammals	35 (9, 72)	102 (26, 257)	11 (8, 46)	4 (0, 10)
Fish	61 (15, 86)	98 (67, 147)	49 (24, 63)	7 (4, 13)
Meat (traditional)	15 (5, 29)	29 (15, 63)	14 (7, 23)	1 (0, 5)
Imported foods				
Meat (imported)	194 (97, 253)	70 (31, 152)	143 (74, 193)	164 (110, 248)
Fruit/Fruit juice	117 (35, 176)	40 (13, 200)	99 (37, 146)	80 (51, 153)
Vegetables	202 (132, 277)	83 (37, 264)	192 (142, 257)	176 (135, 241)
Dairy products	153 (40, 242)	83 (18, 171)	74 (35, 178)	140 (57, 181)
Cereal products	236 (165, 330)	198 (53, 260)	236 (186, 315)	234 (193, 290)
Ultra-processed, imported foods				
Cake	4 (2, 16)	4 (2, 7)	7 (2, 16)	8 (7, 19)
Candy	7 (2, 22)	2 (1, 12)	6 (2, 9)	7 (3, 17)
Sugar-sweetened beverages	201 (44, 330)	45 (16, 302)	254 (151, 420)	232 (160, 421)
Sugar in coffee/tea	2.0 (0.0, 10.0)	0.5 (0.0, 5.5)	3.5 (0.0, 10.5)	2.0 (0.0, 10.5)
Ultra-processed high-fat foods	20 (4, 31)	12 (1, 24)	17 (7, 23)	23 (9, 32)

Food intake is presented at baseline and during the intervention as median (IQR) grams/day for intake stratified by diet group for different types of foods.

## Data Availability

The datasets generated and/or analysed during the current study are not publicly available but are available from the corresponding author on reasonable request.

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
