# Peer review of "Food Reward after a Traditional Inuit or a Westernised Diet in an Inuit Population in Greenland"

_nutrients, 2022, doi:10.3390/nu14030561_

Round 1

Reviewer 1 Report

The authors compare the differences in food reward between traditional Inuit (TID) and Western diets (WD) after a four-week intervention period. Various measurements on the ‘liking’and ‘wanting’ of food preferences using an established Leeds Food Preferences Questionnaire (LFPQ) and on the food intake using food frequency questionnaire are collected. The positive features of this the paper are the first study using LFPQ performed in the Arctic Inuit population. The lack of supporting data that highlight how food choices impact the changing food environment in the Arctic is a drawback to the paper. Overall, the manuscript offers an interesting perspective on the evolving food environment in Greenland. The manuscript will give a more robust description on the impact of changing food environment by including contribution of food choices and socioeconomic status that was previously described by the authors (Jørgensen, M.E, et. al, 2012; Tvermosegaard, M., et. al., 2015) on the increasing risk of developing cardiovascular disease.

Major point:

  1. In Statistical method section, p. 5, line 219-221, the authors state that they are not sure if the sample size was sufficient to detect any differences between TID and WD groups. There is a concern that the no overall difference observed in the study is due to inadequate sample size.
  2. In Discussion section, p. 10, line 322-324, the cited reference state that preloading sweet shifts the food preferences of the study participants. This raises a concern that the food reward data is confounded by the sweet preloading.

Minor points

  1. The abbreviations for traditional Inuit diet, Westernized diet, Leeds Food Preference Questionnaire, food frequency questionnaire, oral glycose tolerance test should be done once when the words are first used in the document.
  2. When the authors refer to traditional Inuit diet, the letter T in the word “traditional” does not need to be capitalized.

Reviewer 2 Report

Dear Authors, Dear Editor,

There are few research papers on the nutrition of populations from such remote regions, so the title of the manuscript seems promising.  However, I have some questions and comments for the authors, the use of which will hopefully improve the scientific quality of the reviewed work.

1. I suggest presenting in the introduction the current dietary pattern among the Greenlandic population, the nutritional status and the prevalence rates of diet-related diseases.

The information on this subject is fragmentary: some information on dietary energy supply is given (L. 55-58), attention is drawn to the potential occurrence of such diseases (L. 83-84) and at the same time to the high prevalence of obesity, diabetes and CVD (L. 347-349), the data in Table 1 show that the study participants were overweight. Certainly, the provisioning difficulties in Qaanaaq city give some idea of food shortages, as does the finding that rye bread, potatoes, vegetables, apples and lamb are the most preferred foods (L. 352-353), but they certainly do not justify the information about high level of food insecurity (L. 393), which implies hunger.

2. I suggest drawing a diagram of the study, as presenting the material and methods used in the manuscript against the background of a larger study causes confusion, especially in subsection 2.2. I assume this large study was conducted between Sept. 2019 - March 2020, but I guess you can indicate when the subgroup of 44 participants in the food reward study participated? - this question relates to the sentence in L. 115-118: most of these 44 participants or most of the large study?

It is also worth explaining why the subgroup studied included residents of - in my view - extremely different cities, the tiny and northernmost Qaanaaq and a city from the south-west coast, the most urbanised. If this was a deliberate choice, then a further suggestion is to include an additional characteristic, i.e. place of residence, in Table 1.  How can the mature age of the study participants (60 and 55 years) be explained, especially since the authors treated this variable as a limitation of the study (L. 393-394).

3. The Discussion should include a nutritional and health assessment of the changes in food intake shown in Table 2 and the changes in some of the characteristics from Table 1. Which post-intervention diet compares favourably in this respect? Do these changes reflect the dietary guidelines for Inuit population?

Minor comments:

L. 49 - did food availability in Europe actually increase already in the middle of the 20th century (1950s)? (EEC agricultural policy measures were only taken in the 1960s).

L. 50 - Today? - the quoted publication is from 20 years ago.

L. 200 - state what conversion factors were used (literature source)

L. 275 - "after the intervention" or as in Table 2 - "during the intervention"?

L. 316-317 - the indicated percentages of dietary energy are contrastingly different from the results in Table 2, where very low intakes of sugar with coffee/tea and very high intakes from sweetened drinks are shown.

L. 349 - "last centuries" - does this refer to hundreds of years or perhaps years from the turn of the 20th century?

L. 406 - based on the article, one can form an opinion about the Inuit diet, that it is not very varied, based on meat and fat and is poor in vegetables and fruit and fibre due to geographical conditions. There is no mention of the dynamically changing food environment - so why does the article end with this statement?

Kind regards

Reviewer 3 Report

Please find the attached letter.

Round 2

Reviewer 1 Report

This reviewer would like to thank the authors to address the points presented in the previous report and to point out the details missed in the text.

Reviewer 2 Report

Dear Authors, Dear Editor,

The authors generally agreed with my suggestions and made appropriate changes to the manuscript. Thank you for explaining several points and I acknowledge the arguments made. 
I understand the explanations for the text in L. 49-51, but I am not persuaded to write about "modern industrialization" in relation to the mid-20th century. 
I suggest that the word "modern" be replaced with "progressive" in L. 49. 

Kind regards